# Identification and Validation of Nutrient State-Dependent Serum Protein Mediators of Human CD4^+^ T Cell Responsiveness

**DOI:** 10.3390/nu13051492

**Published:** 2021-04-28

**Authors:** Kim Han, Komudi Singh, Matthew J. Rodman, Shahin Hassanzadeh, Yvonne Baumer, Rebecca D. Huffstutler, Jinguo Chen, Julián Candia, Foo Cheung, Katherine E. R. Stagliano, Mehdi Pirooznia, Tiffany M. Powell-Wiley, Michael N. Sack

**Affiliations:** 1Laboratory of Mitochondrial Biology and Metabolism, National Heart, Lung, and Blood Institute, National Institutes of Health, Bethesda, MD 20892, USA; kim.han@nih.gov (K.H.); matthewrodman915@gmail.com (M.J.R.); hassanzs@nhlbi.nih.gov (S.H.); 2Bioinformatics and Computational Core Facility, National Heart, Lung, and Blood Institute, National Institutes of Health, Bethesda, MD 20892, USA; komudi.singh@nih.gov (K.S.); mehdi.pirooznia@nih.gov (M.P.); 3Determinants of Obesity and Cardiovascular Risk, National Heart, Lung, and Blood Institute, National Institutes of Health, Bethesda, MD 20892, USA; yvonne.baumer@nih.gov (Y.B.); tiffany.powell-wiley@nih.gov (T.M.P.-W.); 4Cardiovascular Branch, National Heart, Lung, and Blood Institute, National Institutes of Health, Bethesda, MD 20892, USA; rebecca.huffstutler@nih.gov; 5Center of Human Immunology, National Institute of Allergy and Infectious Diseases, National Institutes of Health, Bethesda, MD 20892, USA; Jinguo.Chen@nih.gov (J.C.); julian.candia@nih.gov (J.C.); foo.cheung@nih.gov (F.C.); staglianoke@mail.nih.gov (K.E.R.S.); 6Laboratory of Human Carcinogenesis, Center for Cancer Research, National Cancer Institute, National Institutes of Health, Bethesda, MD 20892, USA; 7National Institute on Minority Health and Health Disparities, National Institutes of Health, Bethesda, MD 20892, USA

**Keywords:** fasting, refeeding, CD4^+^ T cell activation, SOMAscan, IGFBP1, PYY, IL1RL1, MFGE8, integrative bioinformatics

## Abstract

Intermittent fasting and fasting mimetic diets ameliorate inflammation. Similarly, serum extracted from fasted healthy and asthmatic subjects’ blunt inflammation in vitro, implicating serum components in this immunomodulation. To identify the proteins orchestrating these effects, SOMAScan technology was employed to evaluate serum protein levels in healthy subjects following an overnight, 24-h fast and 3 h after refeeding. Partial least square discriminant analysis identified several serum proteins as potential candidates to confer feeding status immunomodulation. The characterization of recombinant IGFBP1 (elevated following 24 h of fasting) and PYY (elevated following refeeding) in primary human CD4^+^ T cells found that they blunted and induced immune activation, respectively. Furthermore, integrated univariate serum protein analysis compared to RNA-seq analysis from peripheral blood mononuclear cells identified the induction of IL1RL1 and MFGE8 levels in refeeding compared to the 24-h fasting in the same study. Subsequent quantitation of these candidate proteins in lean versus obese individuals identified an inverse regulation of serum levels in the fasted subjects compared to the obese subjects. In parallel, IL1RL1 and MFGE8 supplementation promoted increased CD4^+^ T responsiveness to T cell receptor activation. Together, these data show that caloric load-linked conditions evoke serological protein changes, which in turn confer biological effects on circulating CD4^+^ T cell immune responsiveness.

## 1. Introduction

Caloric restriction, intermittent fasting and time-restricted feeding in animal models [1,2,3,4], in healthy volunteers [5,6,7,8] and in overweight individuals [9] have been found to confer anti-inflammatory effects. Furthermore, caloric restriction, mimetic diets and time-controlled fasting reduce inflammatory markers in human disease [10,11,12]. The complexity of establishing the mechanisms orchestrating these immunomodulatory effects include the difficulty in distinguishing the relative contribution of adipose tissue remodeling with concomitant weight loss [9,13], the role of changes in the microbiome [14,15,16] and/or how and which immune cells [17,18] or organs [19,20] contribute to this regulation.

The mechanisms implicated by these immunomodulatory effects include pathways modified: by ketogenesis [7,16,21], via gut biome short chain fatty acids, via reduced insulin-like growth factor 1 (IGF-1) and protein kinase A (PKA) signaling [7,9,22], by the induction of corticosteroid-signaling [1,16], via regulation of heme oxygenase-1 (HO-1) [23] and through the upregulation of autophagy [13]. Fewer studies have explored the mechanisms underpinning immunomodulatory effects of fasting interventions in humans [6,7,8,9].

Additionally, a pertinent question is whether the beneficial effects of restricting calories is purely due to the reduction of calories or due to the dietary composition [24,25]. Initial investigations suggest that the dietary content may be less impactful than duration of caloric deprivation [24,26]. Hence time-controlled limitation in calories, per se, may evoke the process of hormesis, which, in part, is the concept where a mild, sublethal stress can protect against subsequent excessive stressors [27,28]. This biological phenomenon has been shown where caloric restriction increases resistance to subsequent thermal or oxidative stress injury (reviewed [29]). Interestingly, data is emerging that macrophage polarization may be modulated by hormetic triggers [13,30].

Given the advances in high-throughput ‘omic’ studies [31], a potential avenue to explore mechanisms involved in caloric load-dependent immunomodulation could be via the study of serum proteomics from individuals’ serum extracted under different caloric load conditions. The potential utility of this approach is supported by the finding that serum extracted after time-controlled fasting and refeeding in healthy volunteers and in asthma subjects differentially regulated the NLRP3 inflammasome response when used as the exclusive incubation serum in transformed THP-1 monocytic/macrophage cells [6,12]. A recent study identified that time-controlled fasting similarly blunted CD4^+^ T helper cell immune responsiveness [32]. In that study, RNA-seq and flow-cytometric analysis was performed on peripheral blood mononuclear cells from healthy volunteers in response to a baseline overnight fast, a 24-h fast and 3 h of refeeding. The findings included that the greater duration of the fast was linked to greater effects on blunting immune responsiveness and this fasting effect was very robust in blunting CD4^+^ T helper cell activation [32].

To evaluate the effects of changes in serum proteins in response to fasting and refeeding, serum from the same subjects in response to overnight fasting, 24-h fasting and refeeding [32] were analyzed using SOMAscan proteomics, and multivariate analysis was employed to identify caloric load-dependent candidate proteins. Additional candidates were identified using integrative bioinformatics by combining the SOMAscan results with the previously published peripheral blood mononuclear cell (PBMC) RNA-seq data (GEO database link: https://www.ncbi.nlm.nih.gov/geo/query/acc.cgi?acc=GSE165149, accessed on 20 January 2021). The immunomodulatory role of candidate proteins was then functionally validated in primary CD4^+^ T cells extracted from a group of healthy volunteers and from a validation cohort of lean and obese subjects. In this study, we identified multiple novel nutrient load-dependent circulating proteins that either promote or blunt CD4^+^ T cell responsiveness. Furthermore, integrative bioinformatic analysis can be used to find common circulating proteins with parallel pro-inflammatory effects in serum from subjects after fasting, and in obese versus lean subjects. These findings support that acute or chronic nutrient load conditions can generate parallel circulating signaling molecules to drive CD4^+^ T helper cell responsiveness and identify that these factors confer paracrine effects on immune cell regulatory programs.

## 2. Material and Methods

### 2.1. Study Design and Subjects

This fasting and refeeding pilot study was registered in ClinicalTrials.gov with the registration number NCT02719899 and approved by the National Heart and Lung Institute IRB. Subjects were screened in the ambulatory clinic and signed consent for the protocol prior to undertaking the study (Visit 1). Subjects initiated the study after an overnight fast with blood drawn for the baseline immune response (Baseline—Visit 2). After overnight fasting, they consumed a fixed 500 calorie meal before 8am in the morning and fasted for 24 h except for unrestricted water intake. Following a 24-h fasting blood draw (Fasting—Visit 3), the subjects ate another 500 calorie meal with post-prandial blood draws 3 h later (Refed—Visit 3). The schematic of the blood draw protocol is shown in Appendix A. The subject group consisted of 10 females and 10 males with an age range from 22 to 29 years (means ± SD: 24.35 ± 1.98), 22.0–28.7 BMI range (means ± SD: 24.56 ± 2.03) and had the following race distribution (White/Non-Hispanic, *n* = 8; Asian, *n* = 8; African American, *n* = 3; multiple, *n* = 1). All these subjects had no history of any acute or chronic disease. Subjects had a choice between two isocaloric breakfasts: option (1) vegetable omelet, toast with butter and jelly and orange juice; option (2) oatmeal with walnuts, brown sugar, dried cranberries and milk. Lean and obese African American females with an age range from 24 to 78 years (lean: 53.73 ± 18.49 and obese: 53.00 ± 12.45) and BMI’s of 24.17 ± 2.17 (lean subjects, *n* = 15) and 40.29 ± 8.06 (obese subjects *n* = 15), respectively, were enrolled on the Disease Discovery Natural History protocol (NCT01143454). The blood from healthy volunteers for functional validation of target proteins was obtained from on the same Disease Discovery Natural History protocol (NCT01143454) and from the NIH clinical center blood bank (NCT00001846).

### 2.2. Blood Preparation and Bioassay

For the serum bioassay, blood was allowed to coagulate at room temperature in serum tubes for 1 h and serum was collected by centrifuge for 10 min and stored at −80 °C. Each serum sample was analyzed with each assay kit of IGFBP1 (RayBiotech), Peptide tyrosine tyrosine (PYY) (RayBiotech Peachtree Corners, GA, USA), ApoE (Thermo Scientific, Waltham, MA, USA) and PCSK9, Leptin, MFG-E8 and ST2/IL1RL1 (R&D Systems, Minneapolis, MN, USA) according to the manufacturer’s instructions. The absorbance at each assay was read at a wavelength of 405 nm with a plate reader (BioTek, Winooski, VT, USA). The mean value of each subject represents the average of duplicate experiments (*n* = 18 biological distinct subjects).

### 2.3. PBMC RNAseq Analysis

#### RNA Sequencing and Bioinformatics Analysis

Libraries were prepared using RNA extracted from PBMCs of obese and lean subjects (*n* = 4, in each group) and were sequenced on Illumina Novaseq for paired-end 100 bps using standard Illumina sequencing primers. RNAseq fasta file quality was checked using the FastQC (http://www.bioinformatics.babraham.ac.uk/projects/fastqc, accessed on 12 June 2020). The adapters were trimmed using Trimmomatic [33]. The RNA sequence data was aligned to the human genome (GRCh38) using Spliced Transcripts Alignment to a Reference (STAR) [34]. FeatureCounts was used for gene expression quantification and Limma-voom [35,36] was used to perform differential expression analysis. Genes with *p* value < 0.05 were considered differentially expressed (DE).

### 2.4. Cell Culture and Transfection

Primary PBMCs were isolated from human blood by density centrifugation using a lymphocyte separation medium (MP Biomedicals, Santa Ana, CA, USA). CD4^+^ T cells were negatively selected from PBMCs using the CD4^+^ T cell isolation kit (Miltenyi Biotec, Gaithersburg, MD, USA). Cell purity of more than 95% was obtained after CD4^+^ T cell isolation. For knockdown experiments, Accell control siRNA and SMARTpool Accell siRNAs targeted IGFBP1 or PYY (Dharmacon, Lafayette, CO, USA) and were transfected with Accell siRNA delivery media or T cell nucleofector kit according to manufacturer’s instructions (Lonza, Bend, OR, USA). To increase cell viability, CD4^+^ T cells were transfected with T cell nucleofector kit (Lonza) and maintained in media supplemented with 50 ng/mL IL-2 (Peprotech, Cranbury, NJ, USA) for the first 24 h. The IL-2 media was then replaced with regular RPMI media before the T cell receptor-mediated activation. THP1 human monocytes and H9 human T lymphocytes (derivative of HuT78) obtained from ATCC were maitained in RPMI 1640 media supplemented with 25 mM HEPES, 10% heat-inactivated FBS and penicillin/streptomycin.

### 2.5. Cell Stimulation and Cytokine Assays

CD4^+^ T cells from subjects were activated with plate-coated 5 μg/mL CD3 and 10 μg/mL CD28 (BioLegend, Dedham, MA, USA) for 3 days in the presence of 10% autologous fasted or refed serum. To see the effect of heat-inactivated serum (HI) in CD4^+^ T cells, the serum was incubated at 56 °C for 30 min and the CD4^+^ T cells from subjects after overnight fasting (Baseline) were incubated for 3 days with 10% HI and non-HI serum, respectively. THP-1 cells were differentiated into macrophages by incubation with 5 ng/mL PMA (Sigma-Aldrich, St. Louis, MO, USA) for 48 h in media supplemented with 10% fasted or refed serum from the subjects. THP-1 macrophages were incubated at 5 × 10^5^ cells per well in a 96-well plate with 10 ng/mL LPS for 4 h (Ultrapure *Salmonella minnesota* R595; Enzo Life Sciences) and with 10 μM Nigericin (Sigma-Aldrich) for the last 30 min to induce IL-1β secretion. H9 cells were incubated for 24 h in media supplemented with 10% fasted or refed serum, and then activated at 1 × 10^6^ cells per well in 96-well plate with plate-coated CD3 and CD28 for 24 h to induce IL-2 secretion. CD4^+^ T cells (4 × 10^5^/well in 96-well plate) from healthy volunteers were activated with plate-coated CD3 and CD28 for 3 days with supplementation with each recombinant protein (100 ng/mL IGFBP1, ST2/IL1RL1 and MFGE8 (R&D Systems)), or 20 nM PYY_3-36_ (Peprotech) for the last 24 h of incubation. Supernatants were collected, centrifuged to remove cells and debris and stored at −80 °C. The levels of cytokines, including IL-1β, IL-2, IFNγ, IL-4, IL-5 and IL-17, were measured by ELISA (R&D Systems). Results were normalized to cell number using the CyQuant cell proliferation assay (Invitrogen, Waltham, MA, USA).

### 2.6. SOMAscan Assay and Data Analysis

Proteomic profiles were characterized using the 1.3 k SOMAscan assay (SomaLogic, Inc., Boulder, CO, USA). The basis of SOMAscan is built on the use of a new generation of protein-capture Slow Offrate Modified Aptamer (SOMAmer) reagents [37]. Using these reagents, the SOMAscan assay is able to comparatively evaluate protein abundance in 50 μL of serum. Generated by a technique referred to as the selected evolution of ligands by exponential enrichment (SELEX), the 1.3 k assay consists of 1305 SOMAmer reagents selected against a variety of human proteins (47% secreted proteins, 28% extracellular domains, 25% intracellular proteins) that belong to broad biological subgroups including receptors, kinases, cytokines, proteases, growth factors, protease inhibitors, hormones and structural proteins. SOMAmer reagents are binned into three separate groups according to the expected endogenous abundance of each SOMAmer’s cognate protein in typical human samples. Each SOMAmer reagent exists in only one of the three groupings. Serum samples (including controls) are then diluted into three concentrations (0.005%, 1% and 40%) in order to create separate groups for high-, medium- and low-abundance proteins, respectively. Through this separation, the SOMAscan assay is able to quantify proteins across a dynamic range spanning more than 8 orders of magnitude. The diluted samples are then incubated with the dilution-specific SOMAmers. Runs in the 1.3 k assay were performed semi-automatically with a Tecan Freedom Evo 200 High Throughput System (HTS), which utilizes 96-well plates. The SOMAscan plate design included buffer wells (no sample added), quality control and calibrator samples provided by SomaLogic. Quality control and calibrators were pooled samples composed of the same matrix as the biological samples being measured in the plate. Following standard data normalization procedures [38], raw data were first transformed by hybridization control normalization, which utilizes 12 spike-in SOMAmer controls to remove well-to-well variance in the hybridization process, followed by median signal normalization to remove intraplate variance due to sample-to-sample differences in loading volume, leaks, washing conditions, etc. It should be noted that this was a single-plate study, so the standard inter-plate calibration normalization step was not necessary. An interactive Shiny web tool [39] was used during the quality control process at every step of the data normalization process. The schematic of the SOMAscan assay and data analysis is shown in Appendix A. SOMAmer candidates with >50% missing values were excluded from the analysis. The missing values were imputed with the half of the minimum values measured for the respective SOMAmer proteins. The dataset was analyzed by the partial least square discriminant analysis (PLS-DA) from mixOmics library [40] in R. The variables with variable importance in prediction (VIP) score >1 were reported. The heatmaps were generated in R using ggplot2. Pathway enrichment analysis was done using the blood transcription module [41] and clusterProfiler [42]. Additionally, disease gene enrichment analysis was performed using the DisGenNet R package [43].

### 2.7. Quantitative PCR Analysis

Total RNA was isolated using the Nucleospin RNA kit (Macherey-Nagel, Bethlehem, PA, USA) and cDNA produced using a first-strand synthesis kit (Invitrogen, Waltham, MA, USA). Quantitative real-time PCR was performed using SYBR green PCR master mix (Roche, Indianaplis, IN, USA) and run on Lightcycler 96 systems (Roche). Transcript levels of IGFBP1, IL1RL1, MFGE8, PKIG and AMIGO1 were measured using validated gene-specific primers (Qiagen, Germantown, MD, USA). The primers of PYY were made by Integrated DNA Technologies (forward: 5′-CGGACACGCTTCTTTCCAAAACG-3′; reverse: 5′-TGGTTGGCAGATCTCCCAGGAG-3′). Relative gene expression was quantified by normalizing Ct values with 18S using the 2^−∆Ct^ cycle threshold method.

### 2.8. Statistical Analysis

Statistical analyses were performed using PRISM (GraphPad Software, San Diego, CA, USA) and R (URL https://www.R-project.org/, accessed on 17 September 2020). For in vivo and in vitro studies, *n* represents the number of biological replicates per group and is reported in the figure legends. For histograms, the means ± SEM for the indicated number of observations are reported. The box plots show the median and upper/lower quartile of the observation and show the mean as ‘+’. The whiskers show Tukey distribution and the outlier levels are shown as individual points. Statistical significance between the two groups was determined using a two-tailed Student’s *t*-test when analyzing the response between groups. A *p* value < 0.05 was considered statistically significant.

## 3. Results

### 3.1. Validation of Paracrine Effects of Fasting and Refeeding Serum on Immune Cell Responsiveness

As previously described, a clinical study was performed on peripheral blood mononuclear cells (PBMCs) to compare cellular gene expression profiles in response to an overnight fast (Baseline), to a 24-h fast (Fasting) and in response to 3 h of refeeding (Refed; Appendix A). In this study, the 24-h fast was shown to have more robust gene regulatory responses versus refeeding when compared to the overnight fast, supporting the concept that the hormetic stress of the longer fast initiates more robust immunomodulatory effects. The unbiased bioinformatic analysis of the data in the PBMCs supported the results that fasting had a robust effect on CD4^+^ T cells, although the study could not distinguish between cell intrinsic regulatory effects versus paracrine effects on the cells from the 24-h fast [32]. Prior studies showed that serum extracted from fasting versus refed subjects and used for the incubation of THP-1 macrophages showed differential modulation of the NLRP3 inflammasome [6,12], although what factors in serum orchestrated this immunomodulation has not been determined. To address this question, study serum from these 21 healthy volunteer subjects was used here to assay whether the serum conferred or contributed to these immunomodulatory effects, and to perform proteomics analysis to identify and validate putative regulatory proteins (Appendix A). To first validate whether these fasting serum effects were operational in THP-1 cells, we assayed the IL-1β released in this macrophage cell line in response to LPS and nigericin, and confirmed that the fasting serum blunted the IL-1β release compared to cells incubated in serum extracted from the same refed study subjects (Figure 1a). We then evaluated whether this serum effect could be extended to T cells by assessing the response of incubation of fasted and refed serum in the human transformed H9 T cell line. Here too, incubation with the fasted serum blunted the release of IL-2, a canonical CD4^+^ T cell-associated cytokine, in response to T cell receptor activation (Figure 1b). To evaluate the effect of this serum on primary human immune cells, CD4^+^ T cells were acquired from study subjects following an overnight fast (Baseline), and then incubated with naïve or heat-inactivated serum extracted from these same subjects. Here too, the exogenous naïve or heat-inactivated serum showed the same pattern, where the 24-h fasting-extracted serum conferred a reduction in both IFNγ and IL-5 release compared to the refed, supplemented cells (Figure 1c).

### 3.2. SomaLogic Serum Proteomic Analysis Identified Candidate Differentially Expressed Peptides from the Three Nutritional States

Serum protein levels were quantified and then subjected to SOMAscan analysis. The data revealed a large number of differentially expressed (DE) proteins in the paired comparisons between the three groups (Wilcoxin rank sum test <0.05, Figure 2a and Appendix A). DE proteins obtained from the three nutrient conditions are depicted in a Venn diagram (Figure 2b). The relative fluorescent intensity of 19 common proteins regulated in three nutrient conditions are shown in a heatmap (Figure 2c). Pathway enrichment analysis using blood transcription modules (BTM) [41] showed that DE proteins from fasting and refed states were enriched in T cell activation (*p* < 0.05, Appendix A) using the ClusterProfiler pathway (*p* < 0.05, Appendix A). We performed a partial least square discriminant analysis (PLS-DA) with all three groups to identify proteins that had the highest variable of importance in prediction scores (VIP score) that could discriminate between the three groups. The PLS-DA plot shows a robust distinction in the serum protein levels in the three nutritional load states (Figure 2d). Overall, 14 proteins had highly robust variables of importance in prediction (VIP) scores (>5), that can discriminate differences in protein levels between the three nutritional groups (Figure 2e). Additionally, disease gene enrichment analysis of the VIPs with score >1 revealed a significant enrichment in heart disease and obesity (Figure 2f).

### 3.3. Enzyme-Linked Immunoassay Analysis Validation of Identified SOMAscan Candidate Peptides

To determine if the levels of the identified VIP score proteins were differentially expressed in the serum from the different nutrient conditions, serum was assayed by ELISA’s targeting six of these proteins. The highest ranked VIP proteins were insulin-like growth factor-binding protein 1 (IGFBP1) and proprotein convertase subtilisin/kexin 9 (PCSK9). Interestingly, the ELISA assays showed the IGFBP1 levels were significantly elevated in the 24-h fasting serum, and the PCSK9 distinguished the baseline compared to the fasting and refeeding states (Figure 3a,b). Interestingly, the leptin levels paralleled PCSK9 with the distinct group being at baseline (Figure 3c). Apolipoprotein E (ApoE) mirrored IGFBP1, with the highest levels in the fasted state, and PYY was most highly expressed in the refed state (Figure 3d,e). The relative fluorescence intensity graph and heatmap measured from the three nutrient conditions is shown in Appendix A.

### 3.4. Characterization of Immunomodulatory Effects of IGFBP1 and PYY

As IGFBP1 and peptide tyrosine-tyrosine (PYY) had high VIP scores, IGFBP1 levels were the most highly differentiated in the fasting group compared to the baseline and refed groups and PYY showed the greatest difference in the refed state compared to the other two nutritional load states, we characterized the effects of their recombinant proteins on T cell activation. Interestingly, PYY was also identified as a highly regulated transcript level in the ClusterProfiler pathway analysis comparing refeeding to both the fasting conditions (Appendix A). Primary human CD4^+^ T cells were studied in response to T cell receptor (TCR) activation and showed that IGFBP1 significantly blunted the secretion of IFNγ, IL-4 and IL-17, and PYY significantly and conversely increased secretion of the same Th1, Th2 and Th17 cytokine levels (Figure 4a and Appendix A). Interestingly, interrogation of publicly available CD4^+^ T cell libraries show that both IGFBP1 and PYY are expressed in activated T cells [44] and we found that these transcripts encoding for these proteins are expressed in CD4^+^ T cells and regulated by TCR activation. Interestingly, RNA expression of IGFBP1 did not show the same pattern as its secreted protein, suggesting post-transcriptional control of IGFBP1 by the nutritional load (Appendix A). To assess the effects of these two proteins, we then employed a siRNA targeting control, IGFBP1 and PYY transcripts in primary CD4^+^ T cells. The transcript levels of IGFBP1 were reduced by ≈55% and PYY by ≈45% (Appendix A). Following the subsequent TCR activation, IGFBP1-depleted cells showed excess secretion of IFNγ, IL-4 and IL-17 levels, whereas knockdown of PYY had the opposite effect (Figure 4b). Together these data support that IGBP1, which is elevated in the fasting state, blunts CD4^+^ T cell responsiveness, and that PYY augments this activation capacity.

### 3.5. Integration of SomaLogic and RNAseq Data to Identify Putative Interacting Pathways in the Modulation of T Cell Responsiveness

In the initial study on this cohort of subjects comparing RNAseq analysis and flow cytometry, it was found that the immunomodulatory effects of the 24-h fast were more robust than the overnight fast when both groups were compared to refeeding [32]. Hence, an additional integrated bioinformatics approach was employed to evaluate if we could uncover additional differences in circulating protein levels that could correlate with PBMC gene expression level changes in response to 24 h of fasting or refeeding. Here, the differentially expressed genes from the 24-h fasting vs. refeeding PBMC RNAseq data (GEO database link: https://www.ncbi.nlm.nih.gov/geo/query/acc.cgi?acc=GSE165149, accessed on 20 January 2021) were correlated with the differentially expressed SOMAmer candidates from the same individuals. This analysis could potentially identify serum protein candidates that may indirectly impact gene expression in immune cells. While correlation is not causation, the serum protein-gene pairs could then be further evaluated in a cell culture to explore mechanisms of immunomodulation. Significantly differentially expressed serum protein candidates identified by univariate analysis were further assessed for correlation with the DE genes from the published RNAseq dataset. Five candidate proteins showed significant correlations, including KLK7 (negative correlation with a subset of 24-h fasting genes), IL1RL1 (positive correlation with a subset of fasting genes), ANGPT1 (negative correlation with a subset of refed genes), BMP10 (negative correlation a subset of refed genes) and MFGE8 (negative correlation with a subset of refed genes). IL1RL1 and MFGE8 were further studied because both of these proteins were linked to immune modulatory effects [45,46], were both significantly induced in the refed compared to the 24-h fasting samples by SomaLogic analysis (Appendix A) and both were confirmed in serum at the protein levels (Figure 5a) and at the transcript level in CD4^+^ T cells (Figure 5b). The functional validation of the effects of these two candidates were validated where recombinant IL1RL1 and MFGE8 induced the secretion of the Th1 and Th17 cytokines IFNγ and IL-17 (Figure 5c).

### 3.6. Evaluation of Identified Serum Protein Candidates in a Lean Versus Obese Subject Cohort

Given the significant enrichment of the VIP serum proteins in obesity, we further evaluated whether any of these serum protein candidates are operational in other nutritional load conditions. We assayed their levels in serum from overnight fasted lean and obese subjects. While the levels of IGFBP1 and PYY were not different between these two groups (data not shown), both IL1RL1 and MFGE8 were significantly elevated in serum and at the transcript levels in CD4^+^ T cells, respectively, in the obese compared to lean serum group (Figure 6a,b). Body mass index from lean and obese subjects is described in Materials and Methods. Although obesity is linked to inflammation through multiple mechanisms [47], in this study exploring CD4^+^ T helper cell responsiveness, the extent of TCR-mediated cytokine production was restricted to Th1 and Th17 cells, as evident by the significantly higher levels of IFNγ and IL-17 in the obese versus lean subject cells (Figure 6c).

To identify potential gene regulation linked to the elevated IL1RL1 and MFGE8, respectively, we intersected the fasting-refed SOMAmer-gene correlation pairs with that of the DE genes from the lean versus obese RNAseq dataset (Appendix A). Specifically, the IL1RL1- and MFGE8-correlated genes from the fasting-refed datasets were used as the discovery cohort. From the refed expression dataset, 4 genes correlated with IL1RL1 and 62 correlated genes with MFGE8. To explore their potential link with obesity, the expression levels of the genes were further analyzed in the lean and obese RNAseq dataset. Notably, the expression levels of 2 genes correlated with IL1RL1, and 11 correlated with MFGE8, which were also differentially expressed in the obese versus lean subjects (Figure 6d and Appendix A). Specifically, the transcript levels of the gene encoding the cAMP-dependent protein kinase inhibitor (PKIG), which correlated with IL1RL1 levels, and of the gene encoding the adhesion molecule with Ig-like domain-1 (AMIGO1), which correlated with MFGE8 levels, showed the same relationship in the lean versus obese subjects (Figure 6e,f). Consistently, the transcript levels of PKIG and AMIGO1 were significantly induced in obese compared to lean subjects (Figure 6g).

## 4. Discussion

In this study, we undertook a serum proteomics analysis using SomaLogic to identify and characterize nutrient-responsive circulating proteins that directly modulate CD4^+^ T cell responsiveness. When exploring the SOMAscan data in isolation and comparing the levels between the three nutritional load states (Baseline; Fasting; Refed), the PLS-DA identified IGFBP1 as an important variable whose levels were mostly highly elevated and differentially regulated by the 24-h fast compared to the other two groups, and conversely, that PYY was most distinctly induced by refeeding. The functional validation studies supported that fasting mediated the blunting of CD4^+^ T cell activation, which was conferred by elevated IGFBP1, and that the increased immune responsiveness of refeeding correlated with increased CD4^+^ T cell activation by PYY. In parallel, univariate analysis of the serum proteins from the fasting versus refed groups identified several proteins that were significantly differentially expressed. Since serum proteins can impart wide-ranging paracrine effects in the body, including on immune cells, we utilized the previously published RNAseq gene expression datasets from the PBMCs of these subjects [32] to specifically explore the correlation in expression between serum proteins and genes. This integration of the RNAseq data with that of the proteomics data delineated several significantly correlated SOMAmer protein-gene pairs. Notably, the strong positive correlations of IL1RL1 and MFGE8 with genes from the refed RNAseq expression dataset further expanded the candidate genes that likely play a role in nutrient load-dependent immunomodulation in coordination with the serum proteins. Consistently, several significantly correlated SOMAmer protein-gene pairs were confirmed by qPCR after supplementing the cell culture media of the CD4^+^ cells with the respective SOMAmer candidate. Additionally, several serum proteins were also significantly enriched in heart disorders and obesity, suggesting their broader role in nutrient load-dependent signaling. Consistent with this, IL1RL1 and MFGE8, which were increased by refeeding, were also elevated in obese compared to lean subjects, and functional validation studies showed that these two proteins specifically augmented Th1 and Th17 cytokines levels in healthy subjects. Taken together, this study found that numerous circulating proteins under different nutritional states contribute to the modulation of the CD4^+^ T cell immune responsiveness by directly or indirectly modulating the gene expression levels, and thus, this study supports the emerging concept of the paracrine role of serum proteins in immunomodulation. Additionally, the concordant upregulation of a subset of these serum proteins and transcripts in both the refed subjects and obese subjects potentially substantiates their broader roles in immunomodulation and metabolism.

Insulin-like growth factor (IGF)-binding proteins (IGFBPs) comprise a family of regulatory proteins that can stimulate or inhibit IGF activity through high-affinity binding [48]. IGFBP1 is a 30-kDa circulating protein that is predominantly expressed in the liver and is regulated by nutrient cues [49,50]. Additionally, its levels inversely reflect the risk of cardiometabolic complications such as atherosclerosis, hypertension and insulin resistance [51]. Its direct role in immunomodulation is less well characterized, although its cognate substrate IGF-1 via AKT-mTOR promotes Th17 cell differentiation [52], and the increased ratio of IGF-1/IGFBP1 is linked to increased activity of monocytes, B, T and NK cells [53]. Interestingly, and consistent with our findings, IGFBP1 levels are known to be induced during fasting [49]. Consistent with the fasting and feeding effect on CD4^+^ T cell responsiveness [32], this study showed that IGFBP1 blunts Th1, Th2 and Th17 cell responsiveness and its genetic depletion in CD4^+^ T cells has the inverse effect. Interestingly, the knockdown effects show that IGFBP1 has cell-intrinsic effects on CD4^+^ T cell immunomodulation. In contrast to the effects with fasting and refeeding, the lack of change in IGFBP1 levels in the lean and obese cohort may reflect the combination of the role of ethnicity in determining IGFBP1 levels [54,55] and the different racial compositions of the fasting/refeeding and the obese/lean cohorts.

Peptide tyrosine-tyrosine (PYY) is a 36-amino acid peptide which belongs to the neuropeptide Y (NPY) family of biologically active peptides, which also includes NPY itself and the pancreatic polypeptide (PP). PYY is produced by the enteroendocrine L cells in the gut and predominantly signals through the G-protein-coupled Y_2_ receptor, although it can also signal through the Y_1_ receptor [56]. The two endogenous forms PYY_1-36_ and PYY_3-36_ are low in the fasted state and released post-prandially [57]. Interestingly, enteral TLR agonists and the butyrate, which are produced by gut bacterial fermentation, increase L cell PYY expression through NF-B-dependent signaling [58]. PYY_3-36_ can also be generated by the cleavage of the amino terminal Tyr-Pro amino terminal residues by the enzyme dipeptidyl peptidase IV (DP-IV). Although the major role of PYY is transduced via binding to Y_2_ receptors in the hypothalamic arcuate nucleus as an anorexigenic input, a limited amount of data implicates its role in immunomodulation. This has been mostly assessed in the myeloid system, where PYY acting via Y_1_ receptors decreased rat peritoneal macrophages’ adhesion capacity and suppressed phagocytosis and NO production in the resident macrophages [59]. Furthermore, an increased inflammatory macrophage response was evidence in the activated NPY PYY double-knockout macrophages [60]. Additionally, the PPY cleavage ectoenzyme DP-IV is present on numerous leucocytes as the surface antigen CD26, and the antigen is expressed on resting T cells and induced during T cell activation. Its role in T cell biology is further supported in that DP-IV inhibitors suppress clonal CD4^+^ T cell responsiveness [61]. The data from this study extend these findings to show that it can also promote CD4^+^ effector T cell responsiveness and that, similarly to IGFBP1, it has cell-intrinsic effects. Interestingly, levels of PYY are blunted with fasting and in response to feeding in African Americans compared to Caucasians [57], and this race distinction may explain the unchanged PYY levels in the lean versus obese African American cohort.

The interleukin 1 receptor like-1 (IL1RL1) encodes the membrane-bound ST2 receptor for IL-33 [62] and is expressed on a subset of T cells and on numerous myeloid cells [63]. In CD4^+^ T cells, the IL-33/ST2 signaling pathway plays an important role in Th2 activation, and interestingly, IL1RL1 variants are linked to increased risk of IL-33-driven type 2 inflammation in asthma [45]. Although the role of the IL-33/ST2 pathway under different nutrient conditions does not appear to have been studied, data show that the Th2 pathway is induced in asthmatic subjects in response to refeeding after a 24-h fast [12].

Milk fat globule EGF/factor VIII (MFGE8), also called lactadhedrin, has two domains including an Arg-Gly-Asp sequence that binds to dendritic cells, macrophage integrins and a phosphatidylserine (PS)-binding sequence, though which it associates with phosphatidyl serine (PS)-containing membranes including plasma membranes and derived exososomes. Interestingly, MFG-E8 is linked to dendritic cell exosome-mediated antigen presentation [64]. At the same time, the genetic depletion of MFG-E8 leads to murine autoimmunity linked to the impaired clearance of germinal center B cells due to the role of MFGE8 in binding to PS on apoptotic B cells to facilitate their engulfment by macrophages [46]. A role of MFGE8 in the modulation of CD4^+^ T cell function does not appear to have been shown previously, and here we showed that it is both nutrient-level dependent and plays a role in the activation of Th1 and Th17 CD4^+^ T cells.

Numerous limitations are present in this study that should be highlighted. Firstly, in the time-controlled fasting and refeeding study, the subjects had different meal options, and whether the meal composition played a role in modulating the refeeding serum protein levels was not determinable because of the small study size when the groups were analyzed separately. Secondly, the fasting/refeeding and the lean/obese cohorts were not matched for age or racial ethnicity. Although this is a study limitation, it also underscores that the IL1RL1 and MFGE8 immunomodulatory effects are operational across different ages, BMIs and ethnic groups under different nutritional conditions. Finally, in the fasting and refed study, the time of the blood draws for fasting and refeeding were not the same, which could introduce a circadian rhythm effect. This concept would need to be explored in a follow up study, when blood could be drawn at the same time on different days to exclude circadian effects on time-controlled fasting and refeeding serum protein levels.

## 5. Conclusions

The recognition that restricting calories by numerous interventions, such as intermittent fasting or time-restricted feeding, have ameliorative effects on inflammation is bringing these interventions to the forefront as a potential therapeutic strategy [65]. Numerous immune cell-intrinsic signaling pathways [65] and effects on immune cell localization [20] linked to nutrient load immunomodulation have been defined, and recent evidence also shows that this regulation can be modulated at the transcriptional level in CD4^+^ T cells [32]. Furthermore, numerous studies have also shown that serum metabolites, in part derived from gut bacteria such as short chain fatty acids, and ketones derived from the liver or from dietary interventions, also regulate immunomodulation [21,66,67]. In this study, we expanded the understanding of the regulatory control nodes in nutrient load-dependent immunomodulation by identifying that circulating proteins, including IGFBP1, PYY, IL1RL1 and MFGE8, can similarly affect nutrient load-dependent CD4^+^ T cell regulation. Of these serum proteins, IGFBP1 and PYY were exclusively differentially expressed in the fasted and refed states, suggesting that many such proteins may impart fasting-mediated immunomodulation irrespective of the BMI of the subjects. However, future studies delineating differential immunomodulation in fasted lean and obese subjects will be required to fully understand the nutrient load-dependent signaling immunomodulatory processes that are impacted by subject BMI. Furthermore, we undertook an approach of integrating serum proteomics expression data with the PBMC gene expression data to decipher the potential effects of the serum proteins on immune cell gene expression. Previous work has revealed that broad gene expression changes in PBMCs, contributing to the fasting-mediated immunomodulation [32]. This study helped to delineate the interplay between the changes in serum protein levels and their potential impact on genes expression in the CD4^+^ subpopulation of PBMCs as additional drivers of nutrient load-dependent immune modulation. The expanding complexity of this control further supports that the ability to regulate immune responsiveness by caloric restriction interventions is controlled at cell autonomous, paracrine and probably systemic levels. This in turn may explain why fasting mimetic therapeutic compounds may not necessarily be able to recapitulate the comprehensive ameliorative effects of caloric restriction interventions on inflammation and/or on other systemic effects [65].

## Figures and Tables

**Figure 1 nutrients-13-01492-f001:**
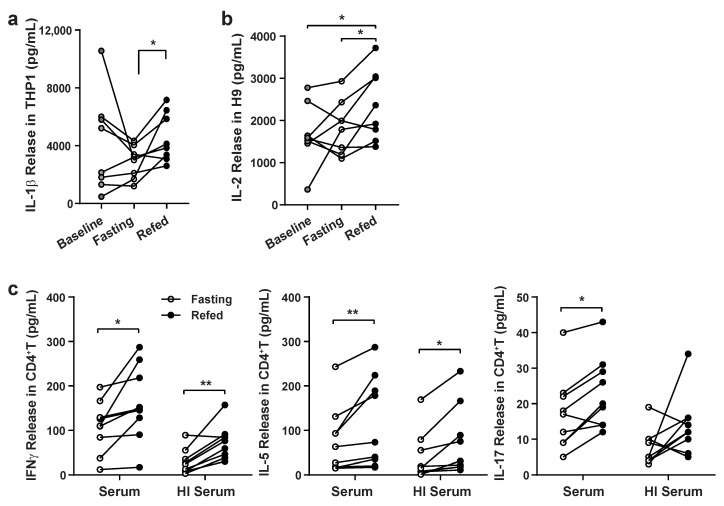
Evaluating whether serum from fasted versus fed subjects has an effect on in-vitro inflammation in cultured cells. (**a**) THP1 cells with 10% serum for preincubation for 1 day and then stimulated with 10 ng/mL LPS for 4 h and 10 μM Nigericin for 30 min. The values represent the average of duplicate experiments (*n* = 8). Paired *t*-test; ** p* value < 0.05. (**b**) H9 cells with 10% serum for preincubation for 1 day and then activated by aCD3/aCD28 for 1 day with 10% serum. The values represent the average of duplicate experiments (*n* = 8). Paired *t*-test; ** p* value < 0.05. (**c**) CD4^+^ T cells isolated from subjects after ON fasting (Baseline) were incubated for 3 days with 10% heat-inactivated (HI) serum, respectively (*n* = 9). HI, 56 °C for 30 min; Paired student *t*-test; ** p* value < 0.05; *** p <* 0.01.

**Figure 2 nutrients-13-01492-f002:**
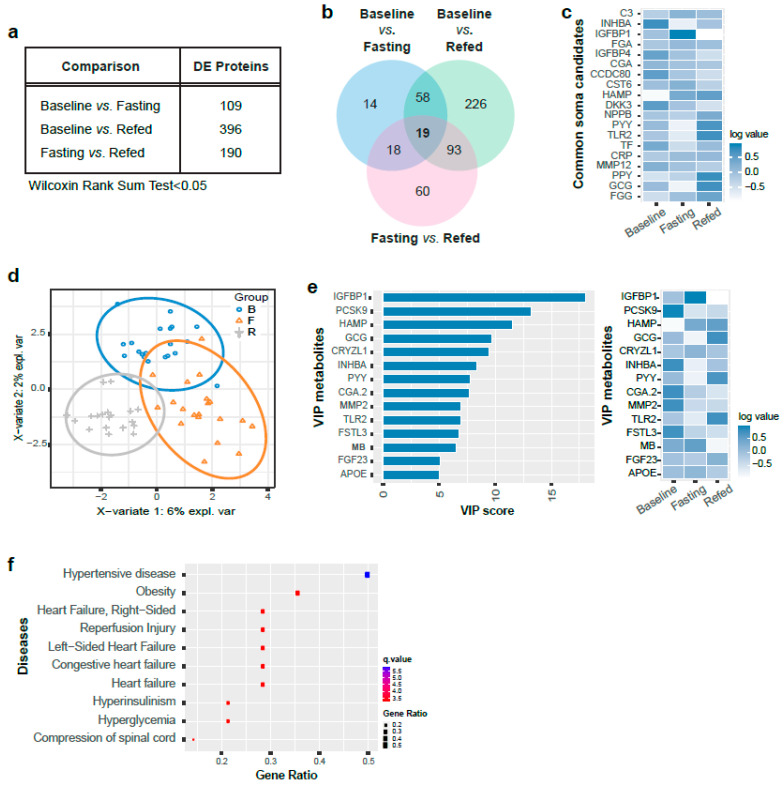
Differentially expressed proteins identified in the indicated comparisons. (**a**) Table showing the number of DE (differentially expressed) proteins identified in the indicated comparisons (Wilcoxin rank sum test <0.05, *n* = 20). (**b**) Venn diagram of DE proteins from three comparisons (Wilcoxin rank sum test <0.05, *n* = 20). (**c**) Heatmap of 19 overlapped proteins identified by SomaLogic analysis in three comparisons (log transformed value of the mean relative fluorescent intensity). (**d**) PLS-DA (artial least square-discriminant analysis) plots from three comparisons (*n* = 20, blue, Baseline; orange, Fasting; gray, Refed). (**e**) Bar graphs of 14 VIP (variables of importance in prediction) proteins from three comparisons (VIP score >5, left panel). VIP scores rank the serum proteins as the most important for differentiating overall serum protein profiles between the groups indicated in the key. All VIP proteins indicated also differed in post hoc analyses (*p* < 0.05). Heatmap of 14 VIP proteins in three comparisons (log value of the relative fluorescent intensity, right panel). (**f**) Disease association analysis with VIP proteins (*p* < 0.05).

**Figure 3 nutrients-13-01492-f003:**
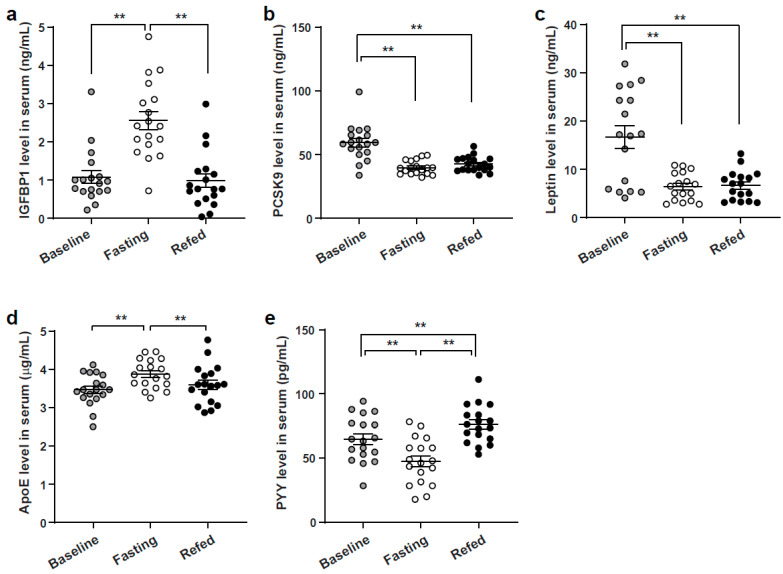
Verify the VIP SOMAscan results by ELISA. (**a**) IGFBP1, (**b**) PCSK9, (**c**) Leptin, (**d**) ApoE, and (**e**) PYY level were measured by ELISA using serum collected from three metabolic conditions (*n* = 18). Paired *t* test; ** *p* < 0.01. VIP, Variables of importance in prediction; IGFBP1, Insulin-like growth factor-binding protein 1; PCSK9, Proprotein convertase subtilisin/kexin 9; ApoE, Apolipoprotein E; PYY, Peptide tyrosine-tyrosine.

**Figure 4 nutrients-13-01492-f004:**
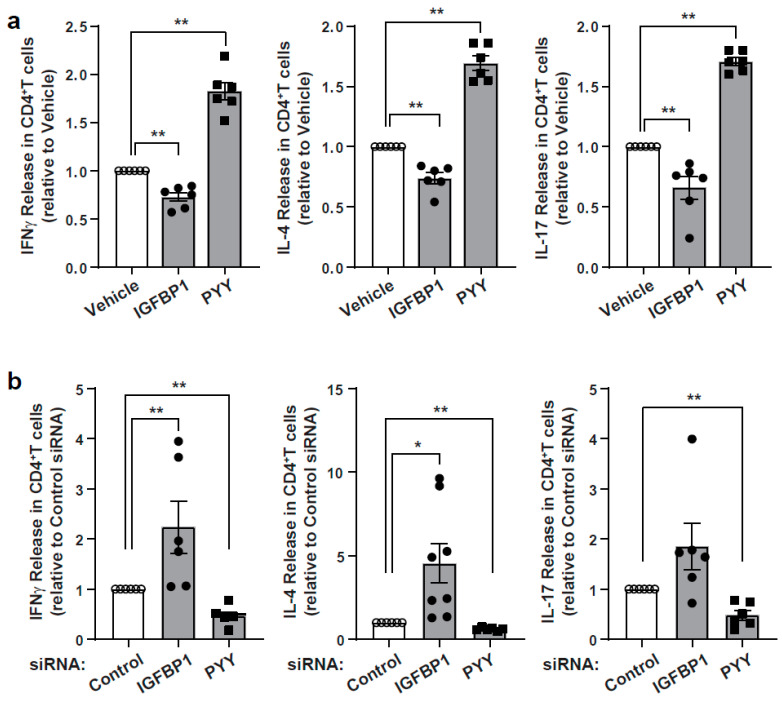
Characterization of immunomodulatory effects of IGFBP1 and PYY. (**a**) Cytokine release after treatment of recombinant proteins (100 ng/mL IGFBP1 or 20 nM PYY_3-36_) for the last 24 h of incubation in CD4^+^ T cells followed TCR activation for 3 days (*n* = 6). (**b**) Cytokine release following KD in CD4^+^ T cells by TCR activation for 3 days (*n* = 6–8). Unpaired student T test; * *p* value < 0.05; ** *p* < 0.01. IGFBP1, Insulin-like growth factor-binding protein 1; PYY, Peptide tyrosine-tyrosine; TCR, T cell receptor.

**Figure 5 nutrients-13-01492-f005:**
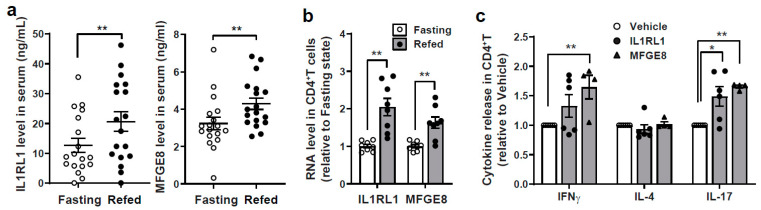
Integrating SomaLogic and RNAseq data. (**a**) Verify the SOMAscan results of IL1RL1 and MFGE8 by ELISA using serum collected from three metabolic conditions (*n* = 18). Paired *t*-test; ** *p* < 0.01. (**b**) RNA expression level in CD4^+^ T cells (*n* = 4–6). Unpaired *t*-test; ** *p* < 0.01. (**c**) Cytokine release after treatment of recombinant proteins (24 h) in CD4^+^ T cells followed by aCD3/aCD28 activation for 3 days (*n* = 4–6). Unpaired *t*-test; * *p* value < 0.05; ** *p* < 0.01. IL1RL1, Interleukin 1 receptor like-1; MFGE8, Milk fat globule EGF/factor VIII.

**Figure 6 nutrients-13-01492-f006:**
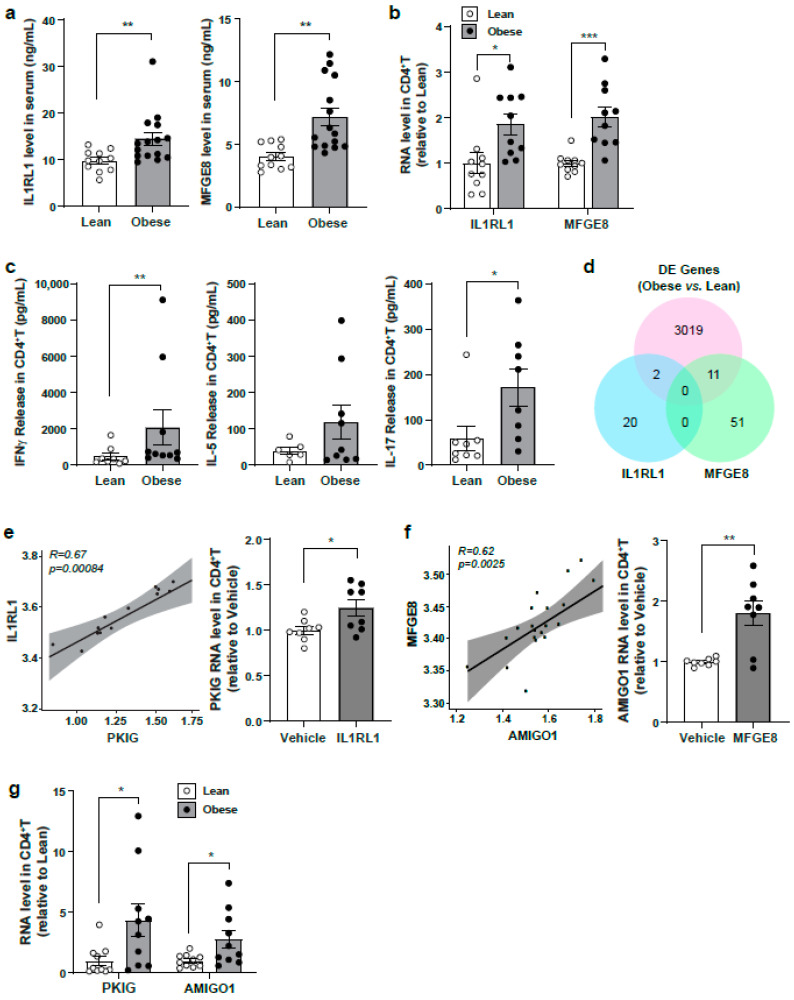
Evaluate the level of fasting serum targets comparing lean to obese subjects. (**a**) IL1RL1 and MFGE8 levels in the serum of lean (*n* = 11) and obese subjects (*n* = 15). Data point of each subject is shown as dot and quantitative data are presented as means ± SEM. Unpaired student *t*-test, ** *p* < 0.01. (**b**) RNA expression in CD4^+^ T cells from lean and obese subjects (*n* = 10). Data point of each subject is shown as dot and quantitative data are presented as means ± SEM. Unpaired student *t*-test, * *p* < 0.05; *** *p* < 0.001. (**c**) Cytokine release in CD4^+^ T cells from lean and obese subjects (lean, *n* = 6–8; obese, *n* = 8–10). Ratio paired Student *t*-test, ** *p* < 0.01. (**d**) IL1RL1- and MFGE8-correlated genes (IL1RL1, 4 genes; MFGE8, 62 genes) from the fasting-refed RNAseq dataset were used as the discovery cohort. The expression levels of these genes were found in the lean-obese RNAseq dataset (IL1RL1, 2 genes; MFGE8, 11 genes), * *p* < 0.05. (**e**,**f**) Integration plot with IL1RL1 or MFGE8 serum protein and correlated genes (left panel) and transcript level correlated with IL1RL1 or MFGE8 in CD4^+^ T cells (right panel, *n* = 8), * *p* < 0.05; ** *p* < 0.01. (**g**) RNA expression in CD4^+^ T cells from lean and obese subjects (lean, *n* = 8; obese, *n* = 10). Unpaired Student *t*-test, * *p* < 0.05.

## Data Availability

The RNAseq data from the fasting and refeeding study is available at the GEO database link: https://www.ncbi.nlm.nih.gov/geo/query/acc.cgi?acc=GSE165149, accessed on 20 January 2021). The differential expressed genes from the obese and lean PBMCs RNAseq dataset are shown in Appendix A.

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
