# Peer review of "Identification and Validation of Nutrient State-Dependent Serum Protein Mediators of Human CD4^+^ T Cell Responsiveness"

_nutrients, 2021, doi:10.3390/nu13051492_

Round 1

Reviewer 1 Report

This work submitted by Kim Han and colleagues is a very interesting study aiming to identify new immunomodulatory serum proteins which could contribute to the anti-inflammatory effects demonstrated in vitro following fasting conditions. They performed a clinical pilot study, collecting sera from baseline, fasting and refed volunteer subjects. Direct measurements were made on these serum as well as several experimental approaches on isolated CD4+ T cells from these sera, including siRNA inhibition studies. This work has been conducted with all the required scientific rigor. By instance, the SOMAscan results are validated by ELISA. The RNA seq results are validated by quantitative PCR analysis. The immunomodulatory properties have been confirmed in vitro in T cells. Further evidence of immunomodulatory potential effects of 2 of the newly identified proteins, Il1RL1 and MFGE8, were further found in lean versus obese subjects. It is well written, very clearly, with all the background, methods, results well described and well presented. The methods which have been used are appropriate and notably the integration of somalogic and RNA seq data allowed the authors to provide very convincing and strong results.

I have only minor comments:

-The levels of IGFBP1 and PYY were found not different between lean and obese subjects (line 354). It would be interesting to have the comments of the authors on this result on the discussion section.

-Please be careful on the typing errors on lines 49,79,360,433.

Reviewer 2 Report

This manuscript investigates proteins that are differentially expressed in the serum of fasted and fed individuals, and the response in terms of cytokine release when T helper cells are treated with these proteins. This paper seems extremely similar to a recently published manuscript from the same group, with a slightly different focus (PYY and IGFBP1 instead of FOXO4 and FKBP5).

Major:

  1. How were the fasting periods decided? Prolonged fasting, used here to describe the 24hour fast, is typically defined as over 24 hours, and assumed to be long enough that subjects would be in ketosis (mentioned in the intro here, but not tested). Why was this fast not longer, where an even greater effect might be seen? Please rephrase—this is not prolonged fasting.
  1. How did the 2 breakfast options differ in terms of macronutrient content, and how many participants chose each breakfast? This is important given recent data that T cell activation can be affected by carbohydrate acting as antigen. How can you ensure the choice of breakfast didn’t affect the cytokine response?
  1. In the obese/lean component of the study, were these subjects fasted or fed, and if fasting, for how long? What did they eat? Why was a racially diverse group with an average age in the 20s compared with an all-female group with an average age in the 50’s of only one race? This makes no sense and potentially confounds the data. Did the lean and obese group have any medical conditions, acute or chronic?
  1. Do the authors have data for fasting that would be at different times of the day? How can you determine the effect seen is from fasting alone vs. circadian rhythm (for example)?
  1. Please explain “For in vivo and in vitro studies, n represents the number of technical replicates per group.”, Line 227-228. I assume this is supposed to read “biological replicates”?
  1. What was the protocol for IDing and eliminating outliers? The statistics section in 2.8 seems far too vague. Was all data normally distributed in order to use Student’s T-test, or were some nonparametric tests used? Were the t-tests paired?
  1. What do the authors take from the fact the PYY and IGFBP1 proteins are no different in lean vs obese subjects? What does this say in light of the fact that these two proteins are supposedly related to nutrient load?
  1. What are the limitations of this study?
  1. More relevant discussion in light of the data presented here is needed. Authors spend a lot of space talking about their 2021 paper, when this should present new findings.

Minor:

  1. What was the FABP1 assay in Section 2.2 used for? Do not recall seeing data for that…
  1. In Section 2.2, authors should state number of technical replicates.
  1. In Line 33 of abstract, add ‘respectively’ to end of sentence ending with ‘activation.’
  1. References from Section 2.3 are not correctly formatted.
  1. Please explain the sentence in Line 151, “To increase cell viability, CD4+ T cells were transfected with T cell Nucleofector kit (Lonza)…”

Round 2

Reviewer 2 Report

Appreciate the responses.